# Carré Phase Shifting Algorithm for Wavelength Scanning Interferometry

**Hussam Muhamedsalih \*** , **Dawei Tang** , **Prashant Kumar and Xiangqian Jiang**

Centre for Precision Technologies, University of Huddersfield, Huddersfield HD1 3DH, UK;
d.tang@hud.ac.uk (D.T.); p.kumar2@hud.ac.uk (P.K.); x.jiang@hud.ac.uk (X.J.)
\* Correspondence: h.muhamedsalih@hud.ac.uk; Tel.: +44(0)-1484-257119

**Abstract:** Wavelength scanning interferometry is an interferometric technique for measuring surface topography without the well-known $2\pi$ phase ambiguity limitation. The measurement accuracy and resolution of this technique depends, among other factors, on the algorithm used to evaluate its sinusoidal interference pattern. The widely used fast Fourier transform analysis experiences problems such as waviness error across the measured surface due to spectral leakage. This paper introduces a new fringe analysis method based on the Carré phase shifting algorithm combined with a least squares fitting approach. Numerical simulation was carried out to assess the performance of the Carré algorithm in comparison to fast Fourier transform analysis, and the same was validated by presenting four experimental case study examples (a surface flat, a ceramic ball bearing, a flexible thin film, and a discontinuous step height sample). The analysis results show that the proposed Carré algorithm with least squares fitting can significantly eliminate the waviness error, especially when measuring steep surfaces.

**Keywords:** wavelength scanning interferometer; Fourier transformer; Carré algorithm; least squares fitting

## 1. Introduction

Ultra-precision modern manufacturing widely fabricates components having micro- and nanostructured functional surfaces such as optical surfaces (e.g., Fresnel lens), silicon wafers, micro/nanoelectromechanical systems (MEMS/NEMS) and flexible thin film electronics [1,2]. The mass production of such components has created a demand for inline inspection tools that are able to perform in situ nanoscale areal measurements that are used as a quality assurance tool and as feedback to control the manufacturing process. Interferometry is widely used for micro/nanoscale surface measurement and has the potential to be used for inline measurement. Examples of interferometer techniques include phase shifting interferometry (PSI) [3], scanning white light interferometry (SWLI) [4], and wavelength scanning interferometry (WSI) [5]. WSI can be employed to measure absolute distances without the $2\pi$ phase ambiguity limitation of PSI and without the mechanical scanning required in SWLI.

The development of wavelength scanning techniques, driven by expanding the scanning range to hundreds of nanometers and linearizing the tuning of the wavenumber (reciprocal of the wavelength) [6], has facilitated the evolution of WSI to the point where it can be employed for micro/nanoscale measurement applications. This evolution has stimulated parallel development of fringe analysis algorithms to achieve nanometer precision without the need of combining WSI with other interferometric techniques. Consequently, many algorithms have been reported for fringe pattern analysis, such as those based on the Fourier theorem, which evaluates the phase slope and the convolution method, which determines the stationary points of the phase change [7–10]. Takeda et al. showed that a periodic fringe pattern can be evaluated by use of Fourier transform and filtering methods, a theorem that has been widely used by other researchers [7,11]. When the interference

signal quality is good, the Fourier fringe analysis (FFA) method provides very accurate results, but as SNR degrades, the accuracy is significantly degraded [12]. WSI is very susceptible to environmental disturbances, and in such a case, the height or surface information estimation using the FFA produces measurement errors. Other factors such as nonlinearity in the wavelength tuning of the light source or harmonic components present in the interference signal due to the multiple reflections and coupling errors are the major sources of systematic errors in the WSI [13,14]. Several phase shifting algorithms have been developed and implemented to compensate for such errors in the measurements and enhance the WSI precision. The precision and accuracy are determined by selection of the algorithm implemented for fringe analysis. For suppressing coupling errors, Miao et al. proposed the 8N-7 phase shifting algorithm consisting of novel polynomial window function and discrete Fourier transform [15]. This 8N-7 algorithm provided the best performance compared to the conventional phase shifting algorithms, but it required a long wavelength range and suppression of the intensity fluctuation and environmental vibrations. Kim et al. proposed another two types of phase detection algorithms using Fourier analysis, which could suppress the harmonic components and the modulation frequency shift in the interferometer signal [16]. Sue et al. proposed a new method random phase shift algorithm with adaptive cavity length to compensate the phase shift from the interferogram sequence [17]. The abovementioned studies have one thing in common, namely the use of FFA for phase calculations. This brings in an inherent leakage effect that originates due to the finite sampling of the interference fringes. An effective way of reducing the spectral leakage is windowing [18]. However, the disadvantage associated with windowing is amplitude compression at both ends, which causes degradation in the quality of phase at ends, especially if the signal is very noisy [19]. Another method is the extension of the boundaries of the fringe pattern using the zero padding, but this makes the process computationally complex and time consuming [20].

Spectral leakage leads to distortions in the phase calculations in the application of the standard filtering process. Zhang et al. proposed a method of improving the measurement error in the WSI by designing and optimizing adaptive passband filters used in the FFA [21]. The measurement results improved, but the associated issues with the finite size of the fringe data remained unsolved. These phase distortions can introduce a waviness error over the measured surface, hence limiting the measurement accuracy and resolution. This paper introduces the implementation of an alternative algorithm to evaluate the phase shift of the wavelength scanning process without the need for the Fourier transform method. This technique is based on combining one of the PSI algorithms, namely the Carré algorithm, with a least squares fitting approach and is referred to as CLS. The results of applying CLS are found to be comparable to measurements obtained from FFA but with the advantage of eliminating the waviness error on the surface.

## 2. WSI Overview

WSI, as shown in Figure 1, can be employed to measure large discontinuous surface topographies with unambiguous height determination. The interferograms are produced with no mechanical movement by scanning the wavelength of a halogen light in the visible region (683.4–590.9 nm in this work) using an acoustic optic tunable filter (AOTF).

Such a measurement methodology can provide significant enhancements in speed compared to conventional offline focal plane scanning interferometers such as SWLI. In addition, the WSI can be stabilized against environmental disturbances by using a built-in control system [22]. The reference interferometer generated by multiplexing the super luminescent diode (SLED) with a filtered wavelength used as a feedback sensor for the control system. A piezoelectric transducer (PZT) moves the reference mirror and is driven by a PI controller to track the alteration in the optical path due to environmental disturbances such as mechanical vibration and refractive index drift. In contrast to single shot PSI, where instant data acquisition is used to overcome the vibration [23], the data acquisition in WSI

is time-variant and, thus, an active stabilization system acquisition method is implemented to compensate for the environmental disturbances.

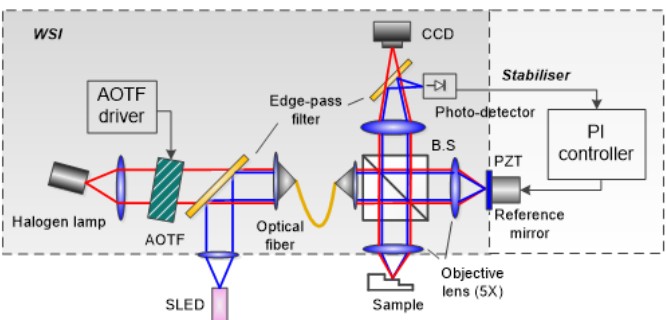

**Figure 1.** Configuration of the WSI.

During the measurement process (scanning the wavelength in the stabilized interferometer), 256 spectral interferograms are captured by a CCD (with sensor resolution of 640 × 480 pixels) over a field of view (FOV) of 5× objective lens (i.e., over 0.57 mm × 0.76 mm), as shown in Figure 2a. A periodic spectral interference pattern, as shown in Figure 2b, can be extracted by registering the intensity values at a given pixel from all the recorded interferograms obtained from wavelength scanning across the wavenumber range to determine the phase shift and, hence, the surface height.

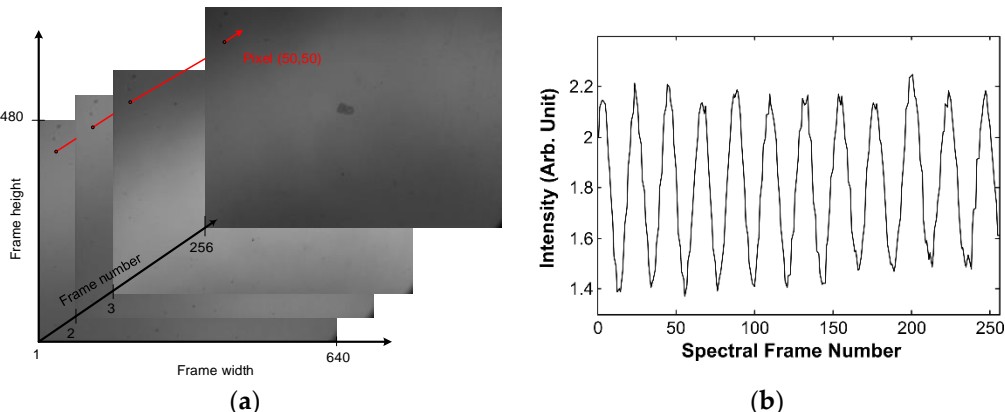

(**a**)                                                                  (**b**)

**Figure 2.** The captured data from the camera: (**a**) the spectral interferograms; (**b**) the interference fringe pattern extracted from one pixel (identified by the red arrow in the interferograms).

The interference pattern of each captured pixel can be expressed by

$$I(x,y;k) = a(x,y;k) + b(x,y;k) \, cos \, [\varphi(x,y;k)] \tag{1}$$

where $I(x,y;k)$ is the interference signal with respect to wavenumber $k$, $a(x,y;k)$ and $b(x,y;k)$ represent the DC background intensity and fringe visibility, respectively. The phase $\varphi(x,y;k)$ is defined by the following formula

$$\varphi(x,y;k) = 4\pi\lambda^{-1} \times h(x,y) + \varphi_0 = 4\pi k \times h(x,y) + \varphi_0 \tag{2}$$

where $h(x,y)$ represents the surface height, $\varphi_0$ is the initial phase.

The sinusoidal signal of each pixel is individually analyzed to determine the height and, thus, a full surface topography within the measurement FOV can be obtained. The flowchart of the analysis is shown in Figure 3.

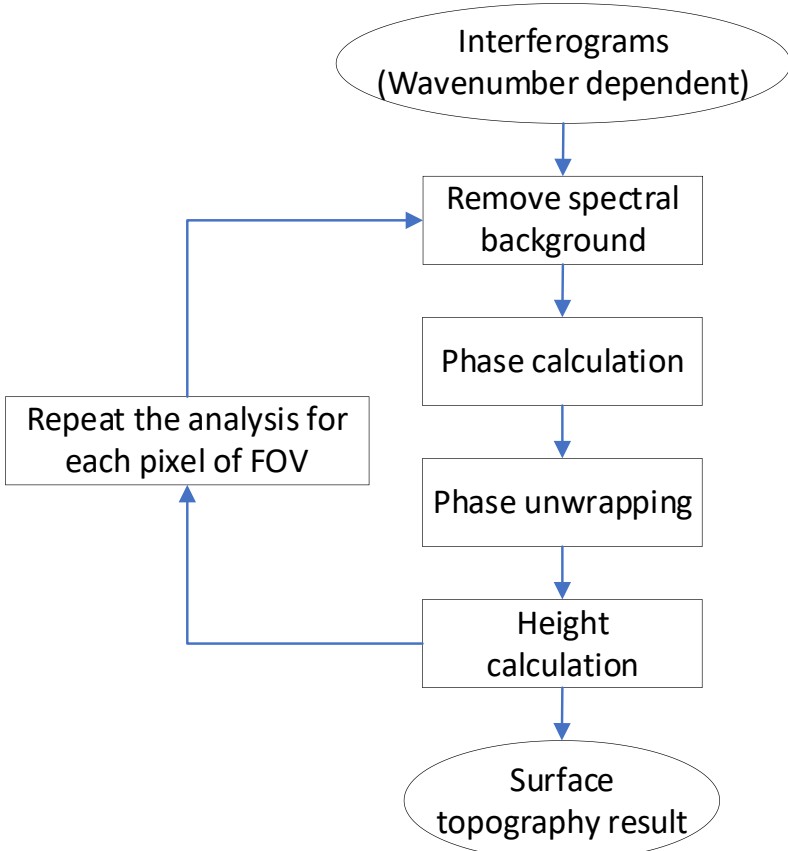

**Figure 3.** The flowchart of fringe analysis for WSI.

The accuracy and resolution of the measurement, among other factors, depends on the algorithm used for fringe analysis. The FFA was reported to be an effective algorithm for WSI [24]. The algorithm procedure is based on extracting the phase information from the unwanted DC bias interference and phase conjugate terms after filtering the spectral power density, which is obtained from discrete Fourier transform (FFT); see Figure 4a. The filtered power density is manipulated by applying the inverse FFT and natural logarithm to find the phase distribution; see Figure 4b. Once the phase distribution has been determined to be a modulo of 2π, the discontinuity jumps are corrected by using a one-dimensional phase unwrapping method. The slope of the distribution is geometrically related to the optical path difference (OPD) (i.e., surface height *h*) as given:

$$h(x,y) = \frac{\Delta\phi}{4\pi\left[\frac{1}{\lambda_{last}} - \frac{1}{\lambda_{first}}\right]} \tag{3}$$

where $\Delta\varphi$ is the phase change related to the scanned wavelength range ($\lambda_{first}$ to $\lambda_{last}$). The distortion at the phase edges, shown in Figure 4c, can result in a measurement error which can be reduced by dropping the distorted data and fitting the phase curve as shown in Figure 4d.

This paper presents an alternative fringe analysis method that can avoid the Fourier theorem and extract a linear (not distorted) phase distribution to enhance the measurement result.

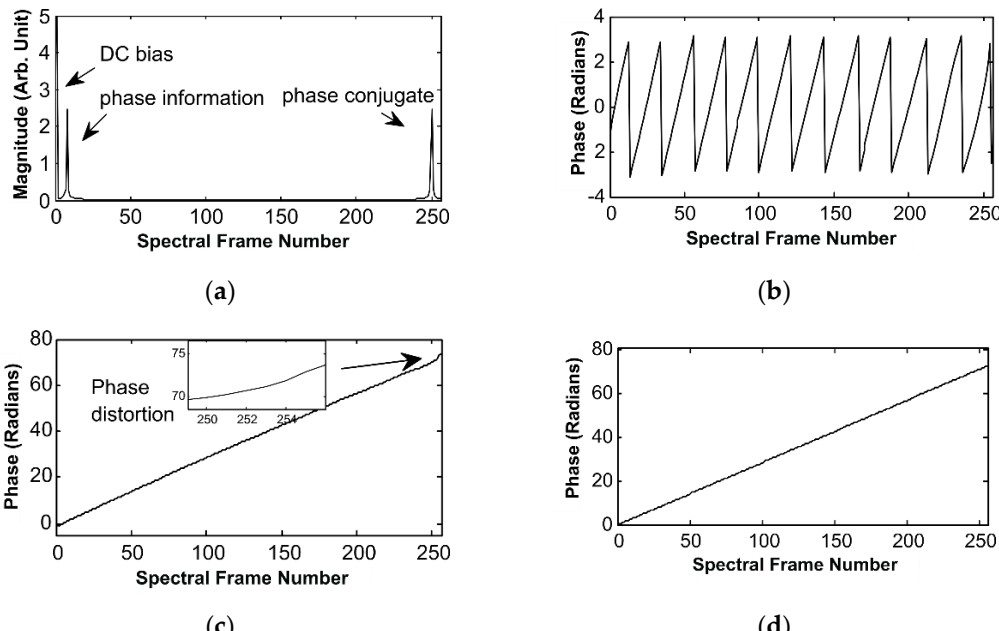

**Figure 4.** Interference pattern analysis using FFA: (**a**) power density of the fringe pattern using FFT; (**b**) wrapped phase; (**c**) unwrapped phase with distortion; (**d**) corrected phase distribution.

### 3. CLS Algorithm

Retrieval of the surface topography information from the periodic interference fringes can be achieved by using a variety of algorithms but, unfortunately, these are associated with limits on accuracy [25]. PSI is well known for its high measurement resolution and accuracy [26]. The precision of phase shifting techniques has driven their associated algorithms to be used for a wide range of interferometers. For example, the algorithms of phase shifting interferometers have been employed to enhance the accuracy of scanning white light interferometry (SWLI) by improving the peak shift detection and fractional phase determination from the position of the best focus frame [27–30]. The classical PSI techniques have also been combined with WSI to improve the measurement precision to better than $\lambda/100$ [26]. Sandoz et. al. [31] showed that the seven-point phase shifting algorithm can be applied on the spectral interferogram for high resolution spectroscopic profilometry. However, the elementary spectral phase shift needs to be increased by $\pi/2$, a shift amount that is difficult to guarantee at different OPD and sampling point in WSI.

In this paper, the phase slope with respect to the wavenumber is determined by using the Carré algorithm and a least squares fitting approach without the need for any mechanical phase shifting mechanism. The phase of the interference pattern obtained by WSI is linearly shifted at each wavenumber [32] and, thus, the intensity values of the interference pattern can be used to determine the phase at a given scanned wavelength. The Carré algorithm was chosen from among other PSI algorithms because the technique is independent of the amount of phase shift ($\alpha$), which is unknown for WSI at a given OPD. However, the amount of the shift needs to be constant at each step, a condition that is satisfied in WSI. The principle of Carré was described in the early literature [33] and can be summarized as showing that the phase can be extracted by modulating and registering the fringe intensity at four equally shifted positions as described in Equation (4).

$$
\begin{aligned}
I_1(x,y;k) &= a(x,y;k) + b(x,y;k) \, cos \, [\varphi(x,y;k) - 3\alpha/2] \\
I_2(x,y;k) &= a(x,y;k) + b(x,y;k) \, cos \, [\varphi(x,y;k) - \alpha/2] \\
I_3(x,y;k) &= a(x,y;k) + b(x,y;k) \, cos \, [\varphi(x,y;k) + \alpha/2] \\
I_4(x,y;k) &= a(x,y;k) + b(x,y;k) \, cos \, [\varphi(x,y;k) + 3\alpha/2]
\end{aligned}
\tag{4}
$$

Analyzing the four intensity values ($I_1$, $I_2$, $I_3$, and $I_4$) can solve the original phase to be modulo $\pi$ using Equation (5).

$$\tan \phi = \frac{\sqrt{[3(I_2 - I_3) - (I_1 - I_4)][(I_2 + I_3) - (I_1 + I_4)]}}{(I_2 + I_3) - (I_1 + I_4)} \tag{5}$$

The phase can be extended to be modulo $2\pi$ by examining the signs of the numerator and denominator and adjusting the phase accordingly [12]. Applying Carré analysis for continuous wavelength scanning will produce a linear phase distribution.

Prior to applying CLS, the interference pattern is divided by a reference background intensity distribution to remove the unwanted amplitude attenuation caused by the non-uniformity in both the light source spectral density and the CCD pixel response. The algorithm first identifies the amount of phase shift by estimating approximately 1/6 of the sampling population per single interference cycle (i.e., amount of shift = N/6, where N is equal to the total captured frames/number of the interference cycles). Discrete Fourier transform can be used to rapidly determine the integer number of interference cycles by simply finding the peak position of the FFT power density. As such, Figure 5a demonstrates four intensity values (denoted as '+' sign) designated to solve the 'initial' phase value at which the corresponding intensity value is denoted as 'o' sign. These intensities are then processed by Carré equation to calculate the phase value. The process is repeated, after shifting the calculation forward by an adjacent pixel each time, to calculate the succeeding phase values until finding the 'final' phase. This method will produce a phase distribution to be modulo of $2\pi$ as shown in Figure 5b, where the first and last points represent the 'initial' and 'final' phases, respectively. The discontinuities in the phase distribution are unwrapped, as shown in Figure 5c, by adding $2\pi$ values incrementally at the jump positions. The calculation removes extreme points from the distribution because the necessary shifted phases are not available. Least squares fitting is used to eliminate the noise appearing in the distribution, as seen in Figure 5d, hence improving the robustness of the algorithm against error sources. Additionally, the fitting can be used to estimate the extreme phase values to make use of the first and last scanned wavelength when calculating the surface height from the phase slope as given in Equation (3). In contrast to the FFA algorithm, it can be observed that there is no distortion at the edges of the phase distribution and, as such, no 'calculation waviness error' is encountered.

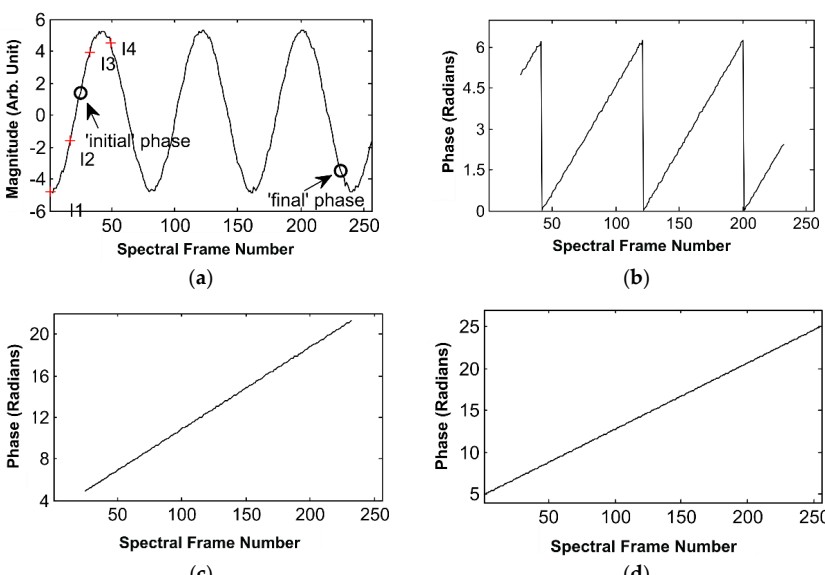

**Figure 5.** Determination of the phase slope using CLS algorithm: (**a**) fringe pattern with shifted phases; (**b**) unwrapped phase; (**c**) wrapped phase; (**d**) least squares fitted phase distribution.

## 4. Numerical Simulation and Measurement Results

The CLS algorithm was verified by simulating interference signals with different surface height values and linear wavenumber across the scanning direction. The patterns were processed using both CLS and FFA method for comparison. To consider the actual measurement conditions, including environmental disturbances and instrument noise, different levels of white noise were added to the interference signals in the simulation. Figure 6 shows the simulation results when the surface heights were varied between 5 and 50 μm and with the noise levels set to either 50 or 25 dB. It can be seen in Figure 6c,d that the FFA method shows similar measurement errors at different noise level because of a filtering window used to suppress the unwanted frequencies. Due to spectrum leakage, there is a noticeable waviness shape in the measurement errors of the FFA method, as can be seen in the Figure 6c. This phenomenon will be more obvious when measuring a steep surface using the FFA method. In contrast, the CLS algorithm can achieve an overall much better measurement accuracy than the FFA approach across the simulated surface heights without noticeable waviness error, thanks to Carre's resolution for determining accurate phase values. Although the CLS algorithm is more sensitive to the noise level due to the nature of the phase shifting algorithms, linear least squares fitting can be used to effectively eliminate phase errors. The measurement error is found to be 0.37 nm (average) and 1.56 nm (maximum) when the noise level is 50 dB, and 6.56 nm (average) and 31.16 nm (maximum) when the noise level is 25 dB.

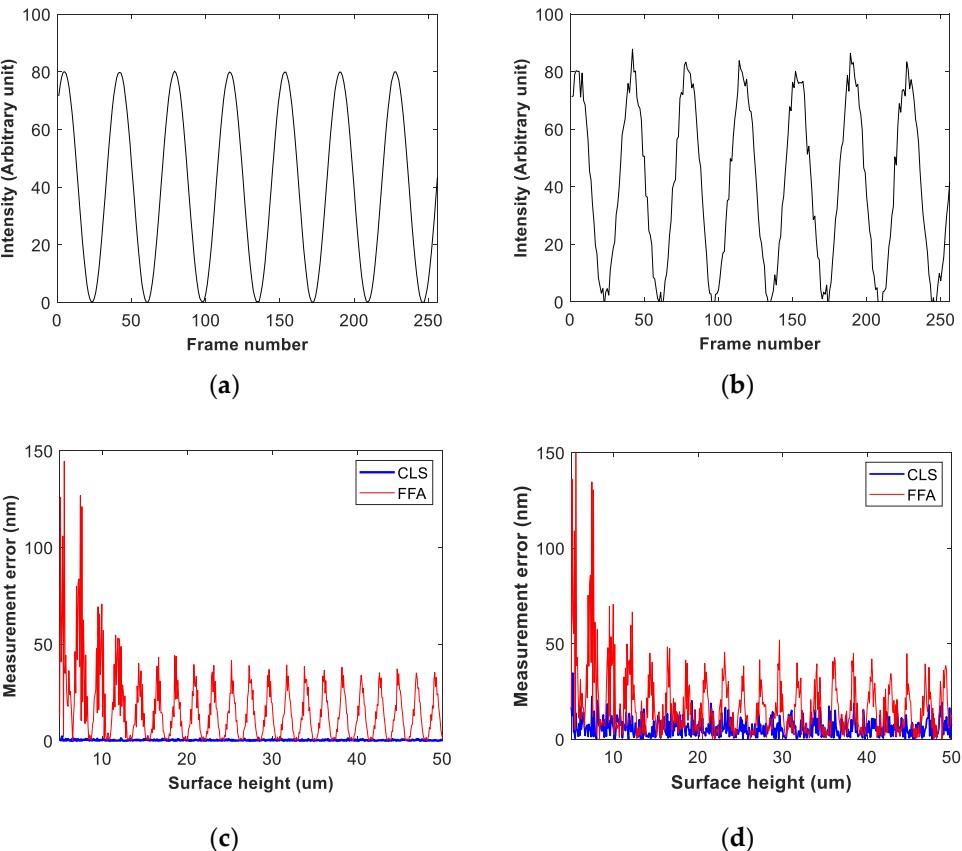

**Figure 6.** Simulation results of measurement errors using CLS and FFA: (**a**) fringe pattern examples with 50 dB white noise; (**b**) fringe pattern with 25 dB white noise (**c**) FFA and CLS measurement errors with 50 dB noise; (**d**) FFA and CLS measurement errors with 25 dB noise.

Additionally, it has been experimentally found that during wavelength scanning, the interference can suffer from unwanted amplitude attenuation. As shown in Figure 7a, we have added some background variations (by reducing term $b(x, y; k)$ in Equation (1) linearly) to attenuate the amplitude of simulated sinusoidal pattern 25% and 50% to inves-

tigate their influence on the measurement accuracy. It was found that CLS still has better accuracy than FFA under different measurement conditions, as shown in Figure 7b. It can be seen that the error decreases as the surface height increases, and there are discontinuities in the error profile at certain surface heights (e.g., 18.4 μm). The error decrease can be resulted from reducing the intensity variation between adjacent pixels, hence reducing Carre algorithm's sensitivity to such phase shift errors. The discontinuity in the error profile can be caused by the amount of phase shift selected automatically by the Fourier transform, as described in Section 3.

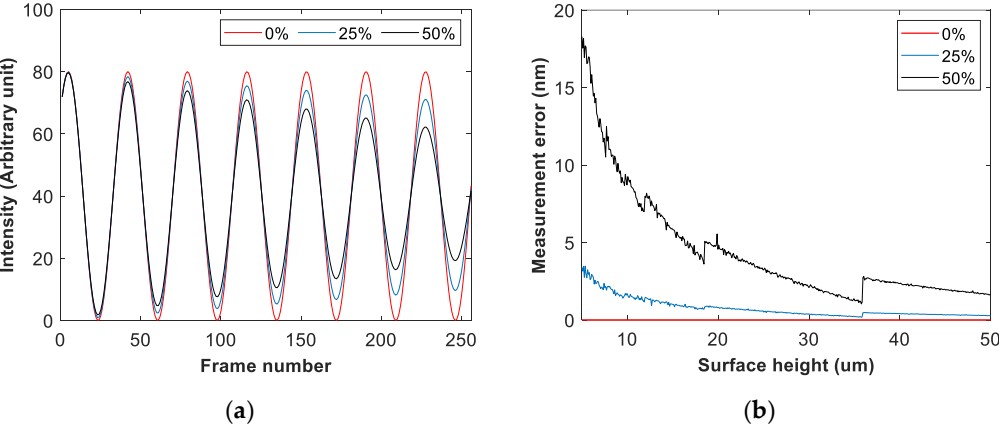

<center>(**a**)         (**b**)</center>

**Figure 7.** Calculated errors due to amplitude attenuation caused by background intensity variation: (**a**) simulated interference patterns examples at different attenuation levels; (**b**) the correspondence CLS measurement errors.

The performance of the CLS algorithm was further investigated by measuring four case study samples using the WSI described in Figure 1, and the obtained results were compared with those for the FFA method.

Case study I: The CLS and FFA were used to analyze the fringe patterns and generate the surface topography, as shown in Figure 8, for the flat surface which has a root mean square (Sq) value of less than 11 nm. While both algorithms can retrieve surface information, the waviness distortion is significant and can be clearly observed in the case of the FFA method. The waviness is found to have the same shape as that of the spatial fringes, which are produced by the gradient of the surface from the reference horizontal plane. As such, the surface roughness of the flat sample was found to be lower using CLS than FFA because of the reduction or the absence of the waviness error, where the Sq for the measured surface was determined to be equal to 9.02 nm for CLS and 24.05 nm for FFA.

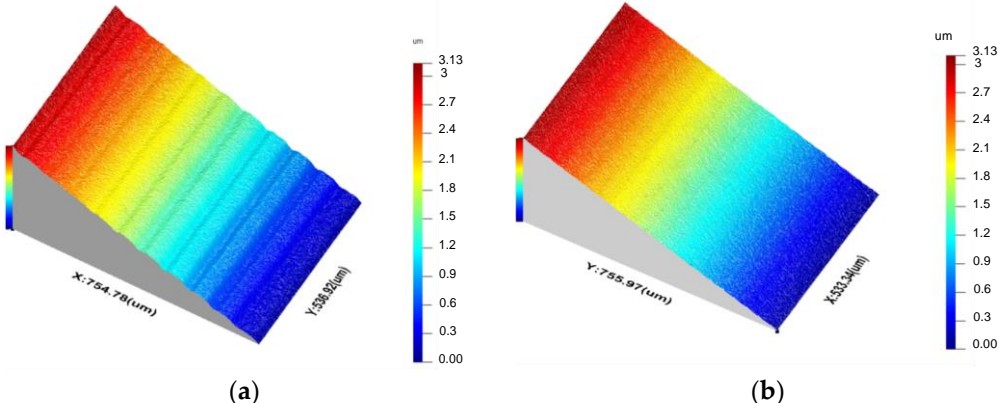

<center>(**a**)         (**b**)</center>

**Figure 8.** Flat measurement using (**a**) FFA; (**b**) CLS.

Case study II: The measurement of the second sample (a ceramic ball bearing) using both algorithms are shown in Figure 9. This figure shows that the surface curvature

experiences periodic discontinuities at certain levels due to spectral leakage in FFA. These discontinuities were eliminated when the CLS method was used.

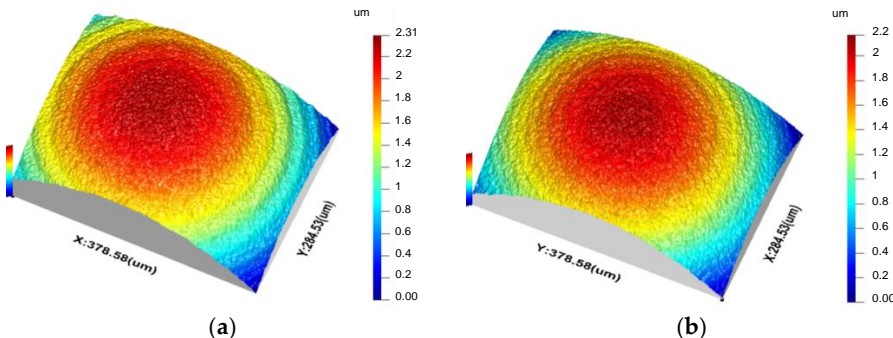

**Figure 9.** Measurement of a ceramic ball using (**a**) FFA; (**b**) CLS.

Case study III: The surface topography of $Al_2O_3$ thin film is shown in Figure 10. The thin film is used as a protective barrier for photo voltaic modules to improve the efficiency by reducing the water vapor ingress [34]. The flexibility of the polymer-based film would usually produce nonflat surface perturbations and, hence, generate multifringes across the FOV. Therefore, the waviness error in FFA could increase the sample roughness compared to CLS, where the Sq was measured to be 32.5 and 21.2 nm, respectively, after levelling (without filtering) the surface.

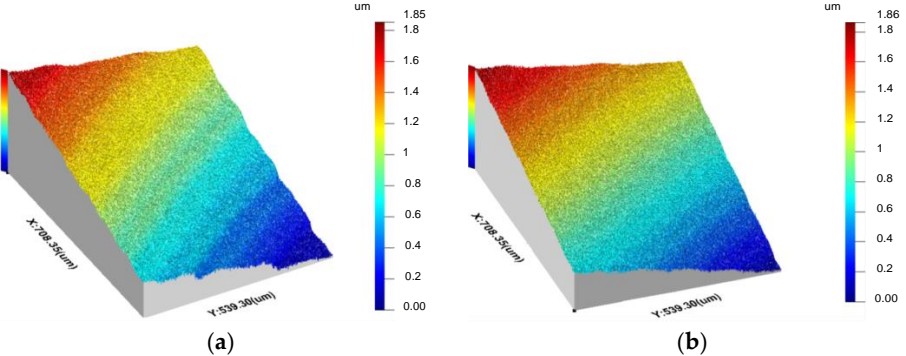

**Figure 10.** Measurement of flexible $Al_2O_3$ thin film surface using (**a**) FFA; (**b**) CLS.

Case study IV: The areal measurement of the fourth sample (a discontinuous surface with steps height of 4.707 µm) is shown in Figure 11. The step height was resolved with good agreement between CLS and FFA as 4.68 and 4.67 µm, respectively; see Figure 12. The spiky errors at the edges of the steps are formed because of the batwing effect, which occurs due to diffraction at the sharp edges or corners of the sample [35]. However, these errors have no influence on the measurement of other pixels because each point is analyzed individually and independently.

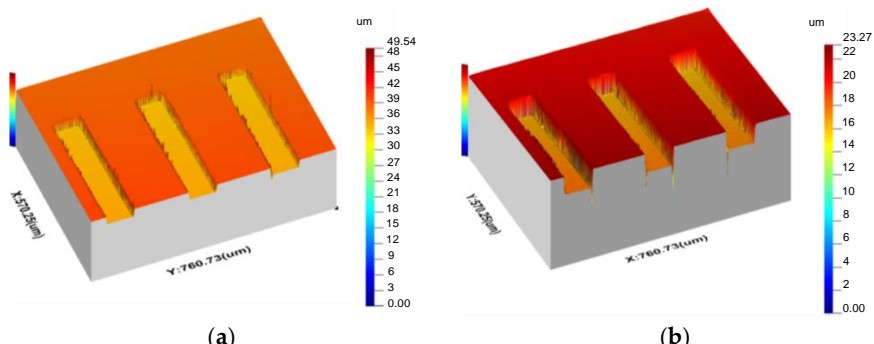

**Figure 11.** Measurement of a 4.707 step height sample using (**a**) FFA; (**b**) CLS.

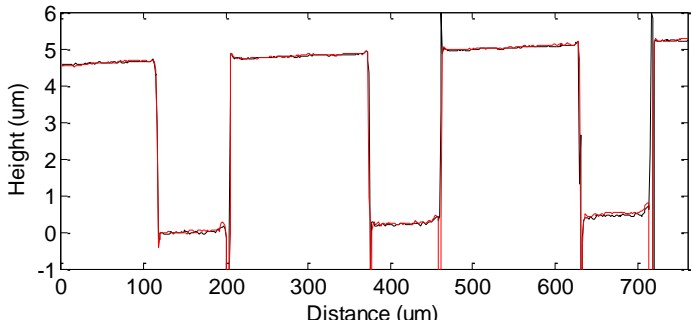

**Figure 12.** Cross-section profile of the step height sample (red is CLS and black is FFA).

**5. Conclusions**

A new algorithm to analyze spectral interferograms has been proposed for application in wavelength scanning interferometry. The Carré phase shift method and least squares fitting (CLS) algorithm can enhance measurement resolution and accuracy at different height levels. In this method, there is no need for spectral filtration and, thus, problems associated with windowing leakage can be avoided. Therefore, the surface slope and curvature can be retrieved with neither waviness error nor discontinuities. Although the CLS needs a phase unwrapping approach, similarly to the Fourier transform method, sufficient sampling points can be used in the described WSI by increasing the number of captured frames.

**Author Contributions:** Conceptualization, H.M. and D.T.; methodology, H.M.; software, H.M. and D.T.; validation, H.M., D.T., and P.K.; formal analysis, H.M.; investigation, H.M. and D.T.; resources, H.M.; data curation, H.M., D.T., and P.K.; writing—original draft preparation, H.M.; writing—review and editing, H.M., D.T., and P.K.; visualization, H.M.; supervision, H.M. and X.J.; project administration, H.M and X.J.; funding acquisition, H.M. All authors have read and agreed to the published version of the manuscript.

**Funding:** This research was funded by Royal Academy of Engineering Industrial fellowship IF2021\108 and RCUK Catapult Researchers in Residency EP/T517732/1.

**Institutional Review Board Statement:** Not applicable.

**Informed Consent Statement:** Not applicable.

**Data Availability Statement:** Data are contained within the article.

**Conflicts of Interest:** The authors declare no conflict of interest.

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
