# Peer review of "Carré Phase Shifting Algorithm for Wavelength Scanning Interferometry"

_machines, doi:10.3390/machines10020116_

Round 1

Reviewer 1 Report

This paper proved the CLS algorithm can significantly eliminate the waviness error, which has some value for engineering applications. Some aspects that should be improved/added to make the paper easier to understand.

  1. why do CLS method can eliminate the waviness error compared to the FFA method, please give more explanation.
    2.What is the relationship between figure 2(a) and 2(b). How is the fringe pattern extracted from one pixel?

3.What can it obtain from figure 2(b)? It is hard to understand the detail and the principle, and need further explanation.

4.Where is the distortion at the phase edges shown in Figure 4c, or should figure 4(c) in line 115 be changed to Figure 4(d)?

5.In Fig.10 the waviness error in FFA could increase the roughness of the sample compared to CLS, but the roughness in fig.9b seems to be higher than fig.9(a), please give more explanation.

  1. What is the reason of irregularity of interference fringes at these points in Fig.11, is it because of the sample itself or the error of your algorithm?

Author Response

Dear, 

First, thank you for your letter and the reviewers’ comments concerning our manuscript entitled “Carré phase shifting algorithm for wavelength scanning interferometer” (ID: machines-1566312). Those comments are all valuable and were very helpful in revising and improving our paper. We have studied comments carefully and have made corrections which we hope to meet with approval. The revised portions have been high lightened in the paper.

The responds to the reviewer’s comments are as follows:

Reviewer 1: Comments and Suggestions for Authors

This paper proved the CLS algorithm can significantly eliminate the waviness error, which has some value for engineering applications. Some aspects that should be improved/added to make the paper easier to understand.

  1. Why do CLS method can eliminate the waviness error compared to the FFA method, please give more explanation.

Author’s Response – The FFA method uses a finite window for filtering the interferometer signal.  Also processing the finite sample data set using discrete Fourier transform produces spectral leakage. And due to this finite widow and spectral leakage the errors in phase extraction are induced which brings in the waviness error. While the phase calculation using the CLS method follows a completely different approach of phase extraction free of spectral leakage issue. Thus, the CLS method provides improved and better results compared to the FFA method.  

  1. What is the relationship between figure 2(a) and 2(b). How is the fringe pattern extracted from one pixel?

Author’s Response - A periodic spectral interference pattern, as shown in Figure 2.b, can be extracted by registering the intensity values at a given pixel from all the recorded interferograms obtained from the wavelength scanning across the wavenumber range to determine the phase shift, and hence the surface height. (The text has been included in the manuscript, lines 107-114)

  1. What can it obtain from figure 2(b)? It is hard to understand the detail and the principle and need further explanation.

Author’s Response – Figure 2.b is the fringe obtained by reading the intensity values at a given pixel from all the recorded spectral interferograms obtained due to the wavelength scanning. So, by processing the fringe at a single pixel will give the phase at that location and consequently the surface height information given by the equation 3. 

  1. Where is the distortion at the phase edges shown in Figure 4c, or should figure 4(c) in line 115 be changed to Figure 4(d)?

Author’s Response – Thanks much for pointing out the mistake. The figure has been revised and has been put in the right order in the manuscript.

  1. In Fig.10 the waviness error in FFA could increase the roughness of the sample compared to CLS, but the roughness in fig.9b seems to be higher than fig.9(a), please give more explanation.

Author’s Response – The roughness in the figure 9(b) looks higher due to the scaling issue in the plot. The scale is varying from 0-2.2 um while in the figure 9(a) the scale extends up to 2.31 um, which is making 9(b) looking like having higher roughness values. This is evident from the other measurements as shown in figure 8 and 10 where the scale values are at similar level.

  1. What is the reason of irregularity of interference fringes at these points in Fig.11, is it because of the sample itself or the error of your algorithm?

Author’s Response – The reason for the irregularity in the step height measurement in the figure 11 is the batwing effect. So, it’s not the sample or the fringes but it’s the batwing effect at the sharp edges in the sample which causes diffraction and so the irregularities or the sharp jumps. A reference [36] has been added in the sentence.

Reviewer 2 Report

Perhaps you should describe the CLS algorithm in more detail. About the abstract: usually it is not recommended to use abbreviations here (WSI, CLS).

Author Response

 Dear,

First, thank you for your letter and the reviewers’ comments concerning our manuscript entitled “Carré phase shifting algorithm for wavelength scanning interferometer” (ID: machines-1566312). Those comments are all valuable and were very helpful in revising and improving our paper. We have studied comments carefully and have made corrections which we hope to meet with approval. The revised portions have been high lightened in the paper.

The responds to the reviewer’s comments are as follows:

Reviewer 2: Comments and Suggestions for Authors

Perhaps you should describe the CLS algorithm in more detail. About the abstract: usually it is not recommended to use abbreviations here (WSI, CLS).

Author’s response – The CLS algorithm has been elaborated in detail, with all the relevant equation in place. All four phase shifted intensity equations have been included and presented in equation 4.

The abbreviations have been omitted form the abstract. (Highlighted)

Reviewer 3 Report

The article presents an innovative algorithm for signal analysis from an interferometer. Despite the relatively simple concept, it seems that the proposed algorithm allows to improve the method and minimize undesirable effects. The work is written in accordance with the requirements of the scientific article. The structure is logical and consistent. The drawback is the lack of references to the latest literature. The article may be interesting both in terms of concept and application. In my opinion, the article may be published after a possible extension of the introduction to include an analysis of the latest publications in this field. 

As I wrote, in my opinion the introduction does not contain references to the latest literature. Authors should supplement the article with a review of publications, especially from the last two years. I do not indicate specific articles, but to my knowledge, at least a few articles on related topics deserve attention. It is not necessary, but it would certainly allow to refer to the authors' work in the context of the latest methods.

Author Response

Dear,

First, thank you for your letter and the reviewers’ comments concerning our manuscript entitled “Carré phase shifting algorithm for wavelength scanning interferometer” (ID: machines-1566312). Those comments are all valuable and were very helpful in revising and improving our paper. We have studied comments carefully and have made corrections which we hope to meet with approval. The revised portions have been high lightened in the paper.

Reviewer 3: Comments and Suggestions for Authors

The article presents an innovative algorithm for signal analysis from an interferometer. Despite the relatively simple concept, it seems that the proposed algorithm allows to improve the method and minimize undesirable effects. The work is written in accordance with the requirements of the scientific article. The structure is logical and consistent. The drawback is the lack of references to the latest literature. The article may be interesting both in terms of concept and application. In my opinion, the article may be published after a possible extension of the introduction to include an analysis of the latest publications in this field. 

As I wrote, in my opinion the introduction does not contain references to the latest literature. Authors should supplement the article with a review of publications, especially from the last two years. I do not indicate specific articles, but to my knowledge, at least a few articles on related topics deserve attention. It is not necessary, but it would certainly allow to refer to the authors' work in the context of the latest methods.

Author’s response - The comments/suggestions by the reviewer has been addressed. Latest publications related to the phase extraction using various phase shifting algorithms for the wavelength scanning/tuning interferometer have been included in the introduction section.  (Lines inserted 45-80, Introduction)

Round 2

Reviewer 1 Report

A  algorithm to analyse the spectral interferogram has been proposed for application in a wavelength scanning interferometer, which is very interesting.